# Autoregressive Video Autoencoder with Decoupled Temporal and Spatial Context

## Abstract

Video autoencoders compress videos into compact latent representations for efficient reconstruction, playing a vital role in enhancing the quality and efficiency of video generation. However, existing video autoencoders often entangle spatial and temporal information, limiting their ability to capture temporal consistency and leading to suboptimal performance. To address this, we propose Autoregressive Video Autoencoder (ARVAE), which compresses and reconstructs each frame conditioned on its predecessor in an autoregressive manner, allowing flexible processing of videos with arbitrary lengths. ARVAE introduces a temporal-spatial decoupled representation that combines downsampled flow field for temporal coherence with spatial relative compensation for newly emerged content, achieving high compression efficiency without information loss. Specifically, the encoder compresses the current and previous frames into the temporal motion and spatial supplement, while the decoder reconstructs the original frame from the latent representations given the preceding frame. A multi-stage training strategy is employed to progressively optimize the model. Extensive experiments demonstrate that ARVAE achieves superior reconstruction quality with extremely lightweight models and small-scale training data. Moreover, evaluations on video generation tasks highlight its strong potential for downstream applications.

## 1 Introduction

Video Autoencoders (Video AEs) typically consist of an encoder and a decoder for dimensionality reduction and video reconstruction. The encoder compresses an input video into a lower-dimensional latent space, capturing the most significant visual features and temporal dependencies. The decoder then reconstructs the video from this latent representation, ideally recovering the original video as closely as possible. With the development of latent generative models in tasks such as video generation and video editing, Video AEs have become increasingly important. By transforming the generation process from the high-dimensional visual space to a compact latent space, Video AEs reduce the computational burden and simplify the learning process.

Traditional Video AEs (Blattmann et al., 2023) split the video into clips and apply neural networks such as 3D attention mechanisms (Islam et al., 2019) that are extended from image AEs to handle the video in segments. However, they model the video with spatial and temporal information intermingled, failing to effectively capture the temporal consistency among the adjacent frames. Indeed, temporal correlations are the defining characteristic of videos, making them crucial for video processing. For two consecutive frames, the contexts in most regions are highly similar with minor variations. On the one hand, incorporating the previous frame for current frame prediction enhances both the single-frame quality and cross-frame continuity. On the other hand, this similarity indicates redundancy, which enables higher compression efficiency when taking into account both the prior and subsequent frames in parallel. To fully leverage the intrinsic flavor of videos, we propose an autoregressive approach, termed Autoregressive Video Autoencoder (ARVAE), to compress and reconstruct the current frame on the basis of the previous one. ARVAE explicitly utilizes the temporal similarity between adjacent frames, and allows for the flexible modeling of arbitrary video lengths. By exploiting the temporality of videos in an autoregressive manner, ARVAE inherits the rich contextual information from preceding frames and only requires a smattering of additional details, promoting the computational saving and video fidelity in downstream applications.

In order to simultaneously achieve the high compression rate and robust reconstruction quality, it is essential to develop a representation that is both compact and informative. In this work, we propose a temporal-spatial decoupled representation that ensures no information loss while maintaining a compact form. To fully leverage the temporal coherence inherent in video sequences, we utilize downsampled motion fields as temporal representation to directly warp the content from the previous frames. To retain complete fidelity of individual frames, we complement each frame with spatial supplement, which provides details that cannot be propagated from the preceding frames. The spatial supplement are computed as the difference between the current frame and the warped content from

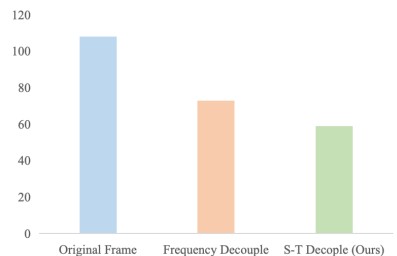

Figure 1: Comparisons of Shannon entropy across different representations.

the previous frame, thereby enabling the recovery of newly emerged content. Moreover, as illustrated in Fig. 1, we analyze the Shannon entropy of various representations on the MCL-JCV (Wang et al., 2016) dataset. Compared with the original frames and the motion representations based on low-frequency and high-frequency decoupling (Wang et al., 2024), our proposed temporal-spatial decoupled representation exhibits low entropy, which is approximately half of the original frames. This result highlights the efficiency of our temporal-spatial decoupled representation, which facilitates more effective compression and network learning (Shwartz-Ziv & Tishby, 2017; Achille & Soatto, 2018).

Specifically, given the previous frame, the encoder compresses the current frame into temporal motion and spatial supplement as latent representations via a temporal encoder and a spatial encoder respectively. The decoder, composed of a temporal decoder and a spatial decoder, reconstructs each frame autoregressively based on its predecessor. To effectively model long-term temporal dependencies, we employ a multi-stage training strategy that gradually increases the temporal length over successive stages. By exposing the model to a wide range of motion magnitudes and sequence lengths, this approach facilitates training stability, accelerates model convergence, and enhances robustness in handling variable-length videos.

Experimental results demonstrate that our method achieves superior reconstruction quality with highly efficient computational costs. Specifically, our approach achieves state-of-the-art performance while reducing training samples by up to 6,700× and model parameters by 80×. Furthermore, the extension of ARVAE to downstream applications further highlights its strong potential for high-performance video generation.

In conclusion, the main contributions of our work are summarized as four-folds:

- We propose a novel approach, namely Autoregressive Video Autoencoder (ARVAE), to compress and reconstruct videos in an autoregressive manner. ARVAE extensively exploits the temporal consistency inherent in video sequences, and enables flexible video processing of arbitrary length.

- We develop an efficient representation by decoupling temporal motion and spatial supplement. This decoupled temporal-spatial representation holds remarkable compression efficiency without loss of information.

- ARVAE autoregressively encodes and decodes the video frame based on its predecessor. A multi-stage training strategy is introduced to promote the training stability and improve the model generalizability.

- The experiments on the benchmarks showcase the effectiveness of our proposed method. ARVAE achieves the state-of-the-art reconstruction quality with $6,700\times$ training data and $80\times$ model parameter reductions. The evaluations on video generation tasks further highlight its potential for downstream applications.

## 2 RELATED WORK

### 2.1 VISUAL AUTOENCODER

Visual latent representation is substantial for existing image/video generation models, which typically projects the input into a latent space and forms compact features for visual synthesis. This technique effectively reduces redundant information of the input and thus saves amounts of computational resources, leadings to remarkable acceleration during inference. Recent development emerges two primary types of representations: discrete and continuous. Discrete VAEs (Van Den Oord et al., 2017; Rombach et al., 2021; Esser et al., 2020; Yu et al., 2024; 2023)compress visual content into discrete tokens by learning a codebook for quantization. However these VAEs have relative poor reconstruction performance and not suitable for training Latent Video Diffusion Models (LVDMs) due to the lack of necessary gradients for backpropagation. In contrast, continuous VAEs (Rombach et al., 2021) compress visual content into continuous latent representations have achieved state-of-the-art visual reconstruction performance. Stable Diffusion (Rombach et al., 2021) was one of the pioneering works to utilize a VAE for image encoding, with a diffusion model then modeling this latent space. This approach greatly reduces image redundancy and facilitating efficient image generation which then become a fundamental step in image generation.

### 2.2 VIDEO COMPRESSION AUTOENCODER

Video compression is a critical challenge in computer vision, there are many methods to expand VAE from 2D image domain to 3D video domain. Several methods extend the latent space from images to videos by incorporated 3D convolutions (Blattmann et al., 2023; Bar-Tal et al., 2024) or spatio-temporal attention mechanisms (Jiang et al., 2024; Peng et al., 2025), which achieves temporal compression on video data. Most recent open-source text to video foundation models (Weijie Kong, 2024; Ma et al., 2025; Wan, 2025) also provide high reconstruction quality VAEs, video generation models were trained base on the latent space provided by those VAEs. However, these VAEs are parameter-heavy and limited to generating fixed-length frame clips at a time, which constrains the flexibility in designing more versatile video generation models. Moreover, the philosophy of decoupling has been employed in traditional video codecs (Le Gall, 1991) for many years, some modern video VAEs (Shen et al., 2024; Jin et al., 2024; Wang et al., 2024; Shi et al., 2024) also borrow this idea for higher compression rate and better reconstruction quality. However, we observe several limitations in existing methods, including the use of large model parameters and the tendency for high compression rates to produce poor reconstruction quality. To address these issues, we propose a lightweight autoregressive video autoencoder that aligns with the inherent autoregressive nature of video data. Our model also employs a decoupled approach to temporal motion and spatial supplementation, effectively reducing video redundancy and improving video reconstruction quality.

## 3 METHODOLOGY

### 3.1 OVERVIEW

Video autoencoders (Video AEs) consist of an encoder that compresses a video into compact latent space and a decoder that reconstructs the video from the latent representation. Different from the previous Video AEs that process video clips in a fixed length, we propose Autoregressive Video Autoencoder (ARVAE) to compress and reconstruct videos frame by frame, which handles a flexible number of frames in an autoregressive manner. The overall framework of ARVAE is illustrated in Fig. 2. By using the outputs of the previous frames, ARVAE encodes the temporal motion and spatial supplement as for the current frame, which is an efficient latent representation that sufficiently leverages the relationship between the neighboring frames. To reconstruct this frame, the overall structure is derived from the last frame through temporal motion, while the spatial supplement completes the detail recovery by incorporating differential contents. For the first frame, we apply an off-the-shelf image-based autoencoder that merely exploits the spatial information.

Specifically, the encoder of ARVAE consists of a motion estimator, a temporal encoder and a spatial encoder. Given the current frame $X_t$ and the predicted last frame $\hat{X}_{t-1}$, the motion estimator calculates the motion field between the two consecutive frames. Then the temporal encoder transforms

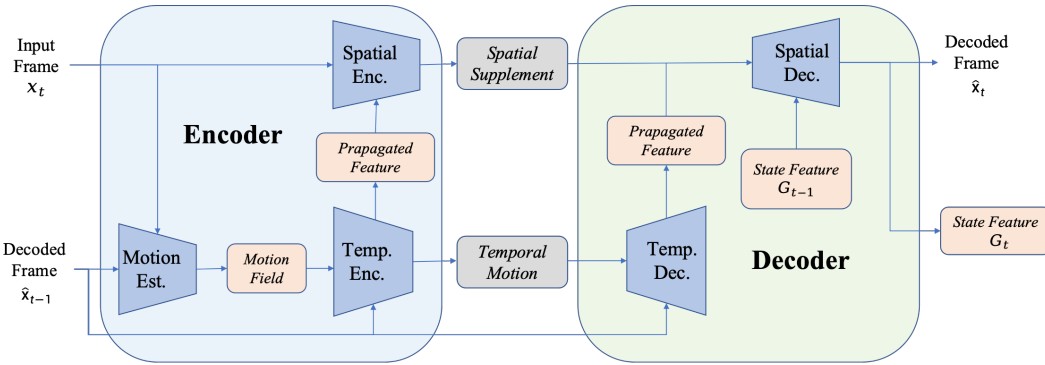

Figure 2: The overall architecture of ARVAE. Based on the previous frame, the encoder compresses the current frame to spatial supplement and temporal motion, and the decoder reconstructs the original frame from the latent representations.

motion field and $\hat{X}_{t-1}$ into the latent temporal motion and multi-scale propagated features simultaneously. The spatial encoder combines the propagated features with $X_t$ to produce spatial supplement, capturing newly emerged content that compensates for missing information from the previous frame.

The decoder of ARVAE consists of a temporal decoder and a spatial decoder. The temporal decoder integrates the temporal motion with the past decoded frame to generate propagated features that represent the global structure of the current frame. The spatial decoder afterwards fuses the spatial supplement with these multi-scale propagated features to reconstruct the original frame.

## 3.2 ENCODE INTO TEMPORAL MOTION AND SPATIAL SUPPLEMENT

Given the last reconstructed frame $\hat{X}_{t-1}$ as reference, our proposed ARVAE compress the current frame $X_t$ into compact latent representations, consisting of temporal motion and spatial supplement that represent inter-frame relationship and intra-frame modification respectively. For a frame of height $H$ and width $W$, the encoder of ARVAE outputs temporal motion $T \in \mathbb{R}^{C_1 \times H/r \times W/r}$ and spatial supplement $S \in \mathbb{R}^{C_2 \times H/r \times W/r}$, where $C_1$ and $C_2$ denote their number of channels and $r$ is the compression ratio.

### 3.2.1 TEMPORAL MOTION

The encoder extracts the temporal motion to capture the inter-frame similarity. Firstly, the motion estimator $f_{ME}$ is applied to predict the motion field $M \in \mathbb{R}^{2 \times H \times W}$ between the last decoded frame $\hat{X}_{t-1}$ and the current frame $X_t$. Then, the temporal encoder $f_{TE}$ takes the motion field and last decoded frame as inputs, and produces temporal motion $T$ and propagated features $P_e$. The motion estimation and temporal encoding process are formulated as follows:

$$M = f_{ME}(X_t, \hat{X}_{t-1}),$$
$$T, P_e = f_{TE}(M, \hat{X}_{t-1}),$$
(1)

As detailed in Fig. 3, the temporal encoder contains three components, including temporal motion extractor, image feature extractor and feature propagation module. The temporal motion extractor processes the motion field to obtain hierarchical motion representations. $N$ motion extraction blocks are applied sequentially, each of which downsamples the motion features by a factor of 2. This results in $N + 1$ stages of multi-scale motion features with resolutions ranging from $[H \times W]$ to $[H/2^N \times W/2^N]$, thereby modeling comprehensive motion representations. The final compact motion features of the smallest resolution are selected as the temporal motion $T$, with a compression ratio $r = 2^N$.

Moreover, the temporal encoder generates propagated features $P_e$ to reflect the information transferable through temporal motion $T$. Similar to the temporal motion extractor, the image feature extractor utilizes a sequence of $N$ feature extraction blocks to acquire the multi-scale pyramid image

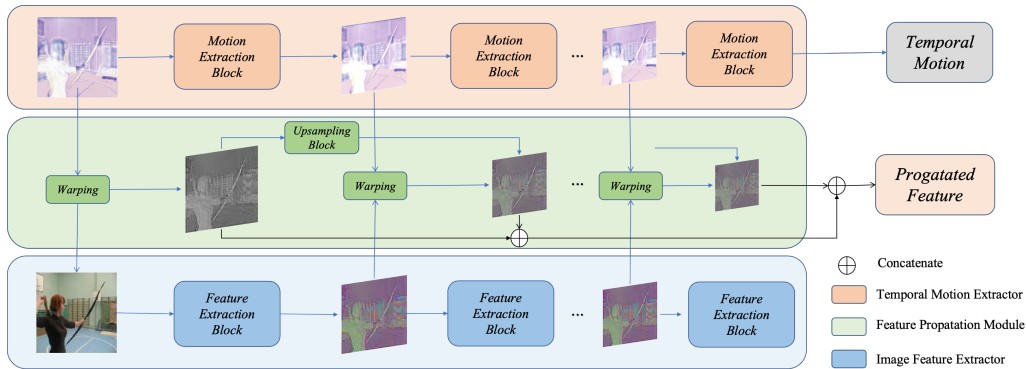

Figure 3: The temporal encoder comprises a temporal motion extractor, an image feature extractor and a feature propagation module, which together generate temporal motion and propagated features. It produces multi-scale visual features and motion representations that are subsequently fused to create a comprehensive representation warped from the previous frame.

features of the previous frame $\hat{X}_{t-1}$. The image features are progressively reduced by half, with spatial resolutions that exactly correspond to those of the motion features. Following that, the feature propagation module fuses the multi-scale motion representations and image features. For each stage, the image feature map is spatially warped via its corresponding motion feature representation to predict the visual representation of the current frame. The transformed features of various spatial sizes then interact across stages. The lower-resolution features are upscaled by a factor of 2 using a upsampling block, aligning the size with that of the adjacent stage. The upscaled features are concatenated with the high-resolution features for enhanced representations. The feature propagation module outputs the updated multi-scale features as propagated features, which contain rich structural information for the current frame derived from temporal motion.

### 3.2.2 SPATIAL SUPPLEMENT

To compensate for contextual information that remains incomplete after temporal propagation, we design a spatial encoder to capture newly emerged visual semantics in the current frame as spatial supplement. The spatial encoder takes the propagated features $P_e$ and the current frame $X_t$ as inputs, and calculates the deviation between them. At the first stage with a resolution of $[H \times W]$, the input image is concatenated with high-resolution propagated features, and is passed through a residual block to produce integrated features that capture spatial differences. The integrated features are then downsampled using a convolutional layer to match the resolution of the propagated features at the next stage. This process is repeated across stages until the final one, resulting in the spatial supplement at resolution $[H/2^i \times W/2^i]$. Therefore, the spatial supplement is obtained through the spatial encoder $f_{SE}$ as follows:

$$S = f_{SE}(P_e, X_t), \tag{2}$$

### 3.3 DECODE LATENT REPRESENTATION TO VIDEO

In an autoregressive manner, the decoder of ARVAE reconstructs the current frame based on the latent representations (*i.e.*, temporal motion and spatial supplement) and the preceding frame. The temporal decoder $f_{TD}$ restores the propagated features by transforming the last frame via temporal motion. The spatial decoder $f_{SD}$ complements the spatial supplement as the residual content to refine missing details.

The temporal decoder shares a similar architecture with the temporal encoder as introduced in Sec. 3.2.1, except for the temporal motion extractor. Different from the temporal encoder that receives the full-resolution motion field $M \in \mathbb{R}^{2 \times H \times W}$ as input, the temporal decoder uses the compact temporal motion $T \in \mathbb{R}^{C_1 \times H/r \times W/r}$ as initial sequential encoding. The temporal motion extractor in the decoder progressively upscales the low-resolution temporal motion to construct multi-scale motion representations, where each motion extraction block upsamples the motion features by a

factor of 2. The image feature extractor and feature propagation module follow the same design as those in the temporal encoder. They produce the propagated features $P_d$ that represent the visual content of the current frame, transferred from the previous frame via motion guidance.

As opposed to the spatial encoder, the spatial decoder progressively upsamples the spatial supplement through convolutional layers and integrates these complementary representations with the propagated features using residual blocks. At the final stage, the propagated features and complementary representations at the full resolution $[H \times W]$ are concatenated with additional state features $G_{t-1}$ from the previous frame, which contain signals that cannot be preserved in visual representations during long-term propagation. A residual block is then applied to these concatenated features to obtain the current state features $G_t$, followed by several convolutional layers that output a high-fidelity reconstruction of the current frame. In summary, the decoder of ARVAE reconstructs the current frame $\hat{X}_t$ as follows:

$$
\begin{aligned}
P_d &= f_{TD}(T, \hat{X}_{t-1}), \\
\hat{X}_t, G_t &= f_{SD}(S, P_d, G_{t-1}),
\end{aligned}
\tag{3}
$$

During training, the recovered frames are supervised by the reconstruction loss combined of three terms, including MSE loss, SSIM loss and LPIPS loss denoted as $L_{\text{MSE}}$, $L_{\text{SSIM}}$ and $L_{\text{LPIPS}}$ respectively, to ensure perceptual and structural similarity of the reconstruction quality. The complete loss function $L$ is calculated as follows:

$$
L = L_{\text{MSE}} + \lambda_1 L_{\text{SSIM}} + \lambda_2 L_{\text{LPIPS}}
\tag{4}
$$

where $\lambda_1 = 0.5$ and $\lambda_2 = 0.5$ in our implementation, trading off between different loss terms.

### 3.4 MULTI-STAGE TRAINING STRATEGY

We adopt a multi-stage training strategy to learn the autoregressive propagation ability of ARVAE by gradually increasing the temporal length over $S$ training stages, enabling the model to generalize to arbitrary length sequences during inference. To avoid the instability of training on long sequences from the outset, we start with three frames in the first stage. This stage allows the model to learn the basic autoregressive propagation mechanism that compresses and reconstructs the current frame based on the previous one. We randomly skip 0–5 frames between adjacent input frames during data sampling, exposing the model to a diverse range of motion magnitudes. However, training only on three-frame sequences leads to error accumulation when propagating over longer horizons. To address this, we progressively extend the sequence length in subsequent training stages, effectively lengthening the temporal context. To prevent overfitting to early frames and to encourage learning on newly added time steps, previously supervised frames are excluded from the loss computation in later stages. Specifically, for the $s$-th stage trained on clips of $L_s$ frames, the overall reconstruction losses are computed and back-propagated from frame $L_{s-1}+1$ to $L_s$. This training strategy facilitates stable training, accelerates convergence, and enhances the model capacity to handle long-term dependencies.

## 4 EXPERIMENTS

### 4.1 EXPERIMENTAL SETUP

**Datasets and Metrics.** During training, we utilize the Vimeo-90K (Xue et al., 2019) Septuplet which contains 91,701 seven-frame sequences with a resolution of $448 \times 256$. Additionally, we incorporate the Inter4K (Stergiou & Poppe, 2022) dataset by randomly sampling 10,000 video clips with a resolution of $256 \times 256$ to further enrich the training data. For evaluation, we employ two standard benchmarks for video compression and reconstruction, including the MCL-JCV (Wang et al., 2016) dataset and WebVid-Val dataset (Bain et al., 2021). The evaluations are conducted on 16-frame videos at a resolution of $256 \times 256$ under three widely-used metrics: Peak Signalto-Noise Ratio (PSNR) (Horé & Ziou, 2010), Structural Similarity Index Measure (SSIM) (Wang et al., 2004), and Learned Perceptual Image Patch Similarity (LPIPS) (Zhang et al., 2018).

**Implementation Details.** We train three model variants with different downsampling settings: $8 \times 8$ spatial downsampling with 4 latent channels ($C_1 = 2$, $C_2 = 2$), and $16 \times 16$ and $32 \times 32$ downsampling with 16 latent channels ($C_1 = 2$, $C_2 = 14$). The training process is divided into three

Table 1: Quantitative comparison with baseline methods. Our model achieves competitive performance with significantly less training data and fewer parameters. # Data and # Ch. denotes number of training data and number of latent channels respectively. ↑ means higher is better and ↓ means lower is better. The best results are highlighted in bold, and the second-best results are underlined.

| Model | # Data | Param. | Down. | # Ch. | MCL-JCV | | | WebVid-Val | | |
|---|---|---|---|---|---|---|---|---|---|---|
| | | | | | PSNR↑ | SSIM↑ | LPIPS↓ | PSNR↑ | SSIM↑ | LPIPS↓ |
| Cosmos-CV | 10M | 101M | 8×8 | 16 | 26.27 | 0.817 | 0.149 | 31.25 | 0.886 | 0.103 |
| CogVideoX | 35M | 206M | 8×8 | 16 | 28.34 | 0.824 | 0.129 | 31.90 | 0.901 | 0.080 |
| VidTok-KL | 16M | 174M | 8×8 | 16 | 29.84 | 0.869 | **0.049** | 33.24 | 0.910 | 0.036 |
| VideoVAE+ | 10M | 500M | 8×8 | 16 | 30.46 | 0.881 | 0.064 | 34.42 | 0.940 | 0.037 |
| ARVAE | 0.1M | 5.9M | 8×8 | 4 | **30.77** | **0.881** | 0.059 | **34.75** | **0.944** | **0.036** |
| Step-Video | 670M | 499M | 16×16 | 64 | 29.73 | **0.875** | 0.073 | 32.50 | 0.905 | 0.054 |
| ARVAE | 0.1M | 6.2M | 16×16 | 16 | **30.12** | 0.870 | **0.066** | **33.40** | **0.910** | **0.053** |
| LTX-Video | N/A | 419M | 32×32 | 128 | 27.46 | 0.787 | 0.146 | 30.25 | 0.853 | 0.093 |
| ARVAE | 0.1M | 6.4M | 32×32 | 16 | **27.88** | **0.793** | **0.145** | **30.55** | **0.873** | **0.084** |

stages, using sequences of 3, 5, and 7 frames respectively. We use the AdamW optimizer (Loshchilov & Hutter, 2017) with a learning rate of $1e-5$, and train for 300K, 210K, 120K steps in each stage respectively. Training us conducted on a single H100 GPU with batch size 64.

**Baselines.** We compare our method with several state-of-the-art Video Autoencoders (Video AEs), categorized into two groups. The first group comprises approaches from prior literature, including Cosmos-CV (Agarwal et al., 2025), VidTok (Tang et al., 2024) and VideoVAE+ (Yazhou Xing, 2025). The second group consists of Video AEs embedded within foundational video generation models, such as CogVideoX (Yang et al., 2024), Step-Video (Ma et al., 2025) and LTX-Video (HaCohen et al., 2024), which typically leverage large-scale training data and substantial model capacity.

### 4.2 COMPARISON WITH STATE-OF-THE-ART APPROACHES

As shown in Table 1, we evaluate our method against baseline models across various compression rates on the MCL-JCV and WebVid-Val datasets. The results demonstrate that our proposed Autoregressive Video Autoencoder (ARVAE) achieves state-of-the-art performance with significantly fewer model parameters and less training data. For the 8×8 spatial downsampling setting, ARVAE achieves high PSNR and SSIM on the MCL-JCV dataset, indicating its ability to reconstruct high-fidelity videos. On the WebVid-Val dataset, our method consistently outperforms all existing approaches, including VideoVAE+ (Yazhou Xing, 2025) that employs a large model.

We also compare our ARVAE with Video AEs provided by foundational video generation models under the 16×16 and 32×32 downsampling settings. In these more challenging scenarios, ARVAE still exhibits superiority over the foundation models, even those trained on extremely large-scale datasets. For instance, ARVAE achieves better reconstruction quality (0.873 vs 0.853 SSIM on WebVid-Val) compared to LTX-Video (HaCohen et al., 2024) at the 32×32 compression rate.

In terms of computational efficiency, ARVAE significantly reduces resource requirements compared to baseline models. For example, Step-Video (Ma et al., 2025) uses 670M training samples and 499M model parameters, while our proposed ARVAE employs only 0.09M videos and 6.2M parameters, resulting in 6,700× and 80× reductions respectively. ARVAE delivers improved performance with resource-efficient configurations, demonstrating both the efficiency and effectiveness of our proposed method.

### 4.3 ABLATION STUDY

**Propagated Features.** To provide rich and comprehensive information propagated from the previous frame to the current one, we design the temporal encoder/decoder to generate hierarchical

Table 2: Ablation study comparing multi-scale propagated features with single-scale propagated features.

| Multi-Scale | PSNR↑ | SSIM↑ | LPIPS↓ |
|:---:|:---:|:---:|:---:|
| ✗ | 24.41 | 0.755 | 0.201 |
| ✓ | 30.12 | 0.870 | 0.066 |

Table 3: Ablation study on the effect of state features. We evaluate image reconstruction performance w/ or w/o state features.

| State Feature | PSNR↑ | SSIM↑ | LPIPS↓ |
|:---:|:---:|:---:|:---:|
| ✗ | 25.08 | 0.836 | 0.176 |
| ✓ | 30.12 | 0.870 | 0.066 |

Table 4: Ablation study on a multi-stage training strategy, progressively increasing the training sequence length from 3 to 5 to 7 frames.

| # Frame | PSNR↑ | SSIM↑ | LPIPS↓ |
|:---:|:---:|:---:|:---:|
| 3 | 28.50 | 0.845 | 0.217 |
| 5 | 29.76 | 0.855 | 0.164 |
| 7 | 30.12 | 0.870 | 0.066 |

Table 5: Ablation study comparing reconstruction quality across different inference sequence lengths.

| # Frame | PSNR↑ | SSIM↑ | LPIPS↓ |
|:---:|:---:|:---:|:---:|
| 4 | 31.12 | 0.890 | 0.065 |
| 8 | 30.62 | 0.879 | 0.067 |
| 16 | 30.12 | 0.870 | 0.066 |

propagated features. To demonstrate the necessity of this multi-scale design, we conduct an ablation study under the $16 \times 16$ downsampling setting. The experimental results in Table 2 show that replacing multi-scale propagated features with a single-scale approach significantly degrades the compression and reconstruction quality. This is primarily because a single-scale design struggles to effectively propagate temporal context through motion dynamics.

**State Features.** For ARVAE that propagates frames in an autoregressive manner, high-quality contextual information is crucial for effective long-term transmission. We analyze that the state features preserve information missing from the visual representations in Table 3, under the $16 \times 16$ compression ratio setting on the MCL-JCV dataset. The experimental results validate the necessity of these state features in maintaining contextual details and temporal consistency during video reconstruction.

**Multi-Stage Training.** We conduct an ablation study to evaluate the effectiveness of our proposed multi-stage training strategy in improving the autoregressive propagation capability of ARVAE. The model is trained on 3-frame sequences in the first stage, and progressively exposed to sequences of 5 and 7 frames in subsequent stages. As illustrated in Table 4, the performance improves significantly with the increase of sequence length. Compared with the single-stage training using three sequential frames, the multi-stage strategy results in higher reconstruction quality (30.12 vs 28.50 PSNR, and 0.870 vs 0.845 SSIM), with fewer perceptual distortions (0.066 vs 0.217 LPIPS). The gradual training on extended temporal contexts enables the model to reduce quality degradation on later frames, effectively capturing long-term dependencies.

**Variable Inference Length.** As shown in Table 5, we evaluate the proposed ARVAE across varying sequence lengths with $16 \times 16$ downsampling on the MCL-JCV dataset. The experimental results show that reconstruction quality gradually declines as the sequence length increases. However, benefiting from our multi-stage training strategy, the performance drop remains marginal, enabling flexible inference over variable-length video sequences.

## 4.4 QUALITATIVE RESULTS

In Fig. 4, we present qualitative comparisons with baseline models under the $8 \times 8$ downsampling configuration, demonstrating the superior reconstruction quality of our approach. Our proposed ARVAE effectively captures fine-grained local details and preserves strong temporal consistency, yielding noticeably clearer results in the later transition frames. These results highlight the effectiveness of our autoregressive design and temporal-spatial decoupled latent representations in maintaining both perceptual detail and temporal coherence. For additional qualitative comparisons, please refer to the appendix and supplementary material.

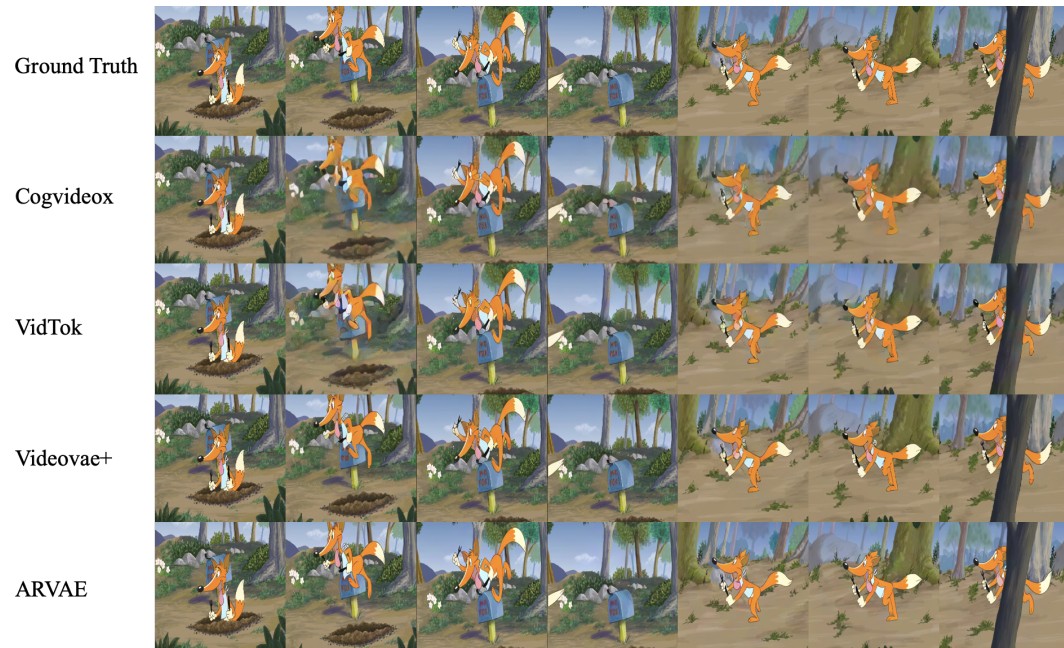

Figure 4: Qualitative comparison with baseline methods. Our model demonstrates the ability to reconstruct fine details and accurately capture rapid motion. The results uses a sampling rate of 8 FPS, and the compression ratio is $8 \times 8$ for all the models.

Table 6: Video generation results across multiple datasets. The evaluations are conducted on 16-frame sequences at $256 \times 256$ resolution, with FVD↓ as evaluation metric.

| Method | Sky Time-lapse | TaiChi-HD | UCF-101 |
|---|---|---|---|
| DIGAN (Yu et al., 2022) | 114.6 | 128.1 | 577 |
| TATS (Ge et al., 2022) | 132.56 | 94.60 | 332 |
| CCVS (Moing et al., 2021) | - | - | 389 |
| ARVAE+SVD | 72.32 | 80.97 | 341 |

### 4.5 APPLICATION TO VIDEO GENERATION

To further validate the effectiveness of ARVAE for downstream applications, we conduct experiments on video generation tasks. Specifically, we follow Stable Video Diffusion (Blattmann et al., 2023) to train a latent diffusion model for synthesizing video latent representations compressed by ARVAE. As shown in Table 6, we evaluate the model on the Sky Time-lapse (Xiong et al., 2018), TaiChi-HD (Siarohin et al., 2019) and UCF-101 (Soomro et al., 2012) datasets, using Fréchet Video Distance (FVD) as the evaluation metric. The results demonstrate ARVAE's ability to generate high-quality and temporally consistent video sequences, highlighting the advantages of its latent representation and its potential for high-performance video generation.

### 5 CONCLUSION

In this paper, we propose a novel Autoregressive Video Autoencoder (ARVAE) approach that encodes and decodes video sequences in an autoregressive manner. ARVAE decouples spatial and temporal information in the latent space, facilitating efficient compression while preserving critical video details. A multi-stage training strategy is employed to handle variable-length sequences. Experimental results demonstrate the effectiveness of ARVAE on both video reconstruction and video generation tasks, with significantly improved computational efficiency.

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

APPENDIX

## A  LLM USAGE

In the context of this conference, Large Language Models (LLMs) are utilized solely as tools to assist with the refinement and enhancement of written communication. Their role is limited to polishing grammar, improving clarity, and streamlining expression. All substantive content, ideas, and conclusions presented remain the original work of the authors. The use of LLMs does not extend to generating research findings, conducting analysis, or producing novel intellectual contributions.

## B  MORE IMPLEMENTATION DETAILS

In the initial stage of training, we use three frames for training. There is an 80% probability that the first and second frames will be used to generate auxiliary features, and a 20% probability that only two frames will be used to reconstruct the second frame.

## C  MORE QUANTITATIVE RESULTS

### C.1  COMPUTATION RESOURCE COMPARSION

Through our decoupling design, we reduce redundancy, resulting in compact latents with a high compression rate. A key advantage of having lower-dimensional latents is the reduced computational resource requirements for downstream tasks. To demonstrate this, we compare the FLOPs and memory consumption of generative models based on representative baselines.

Table 7: FLOPs and memory usage for our model to inference on 1 $16 \times 224 \times 224 \times 3$ resolution videos, and the spatial compression ratio is 16, and channel number is 4.

| Method | FLOPs($\times 10^9$) | #Params |
|---|---|---|
| LTX (HaCohen et al., 2024) | 1086 | 419M |
| Vidtwin (Wang et al., 2024) | 959 | 300M |
| **ARVAE** | 76.8 | 6.2M |

### C.2  SHANNON ENTROPY COMPARISON

In this section, we evaluate the Shannon entropy of latent representations across different models. As previously discussed, lower total entropy indicates a more compact and informative representation. We use VidTok with an $8\times 8$ downsampling ratio as the baseline. To ensure a fair comparison, we compute the total entropy in our ARVAE setup by summing the entropy of the first image and its corresponding latent representation.

Table 8: Shannon Entropy of the latent representation. We evaluate the average entropy of 1000 MCL-JCV 16 frames 256×256 videos.

| Model | Downsample | Channel | Entropy |
|---|---|---|---|
| VidTok | $8\times 8$ | 16 | 292.01 |
| ARVAE | $8\times 8$ | 4 | 44.61 |

## D    MORE QUALITATIVE RESULTS

Ground Truth

StepVideo

ARVAE

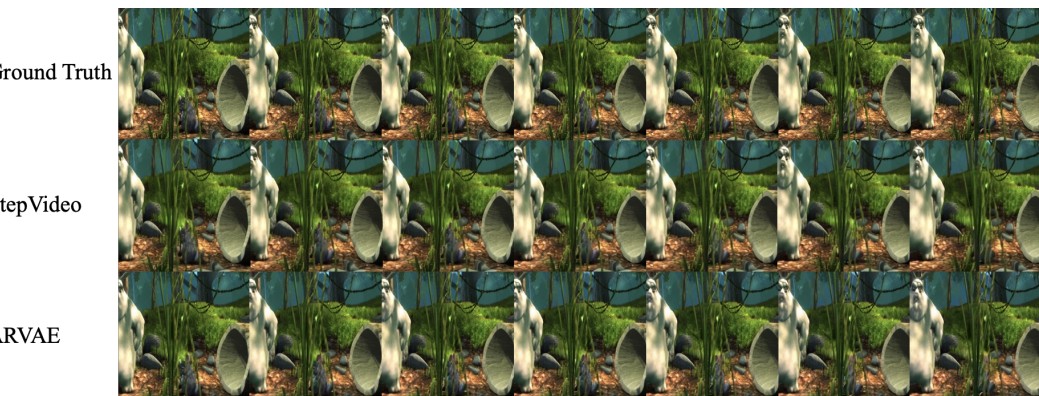

Figure 5: Visualization of $16\times$ compression ratio with 4 channels reconstruction quality. Our model demonstrates comparable reconstruction quality to the autoencoder of open-source large diffusion model.

Ground Truth

LTX

ARVAE

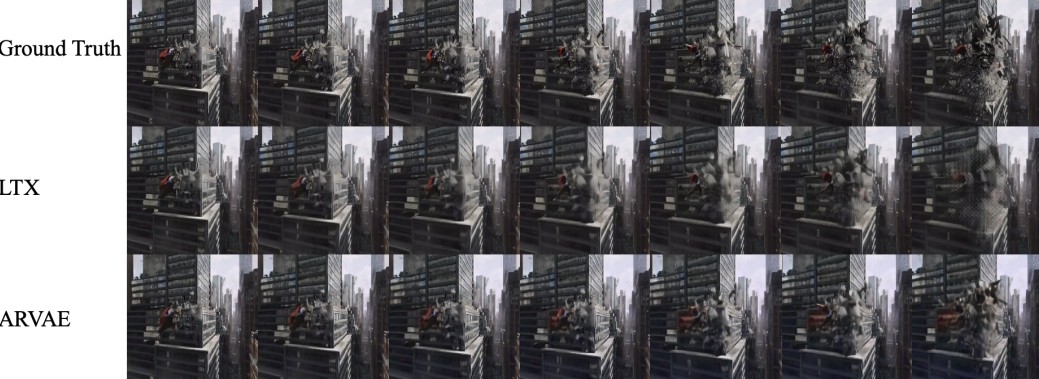

Figure 6: Visualization of $32\times$ compression ratio with 16 channels reconstruction quality. Our model demonstrates the ability to reconstruct fine details and accurately capture transition frame. xw

