# OpenReview forum: "Autoregressive Video Autoencoder with Decoupled Temporal and Spatial Context"
_ICLR.cc/2026/Conference — Submitted to ICLR 2026_

### Official Review · Reviewer_UUbk · 2025-10-29

**Soundness:** 2
**Presentation:** 2
**Contribution:** 1
**Rating:** 2
**Confidence:** 4

**Summary:**

The paper proposed a lightwiehgt autoregressive video autoencoder, referred as ARVAE. Instead of splitting to small video clips or blocks, ARVAE compresses and reconstructs videos in a frame-by-frame way. Given the previous frame, ARVAE calculates the motion field between the consecutive frames with a motion estimator, then the temporal encoder transforms the motion field to temporal motion and multi-scale features, afterwards, the spatial encoder produces the spatial supplement to encode the difference of the current frame. The motion extraction blocks downsample the motion fields in N+1 stages. N feature extraction blocks obtain multi-scale pyramid image after the warping from the previous frame. The spatial encoder applies downsampling and convolution to encode the hierarchical features.  ARVAE has been trained in a multi-stage way. Using very few training videos on Vimeo-90K and Inter4K, ARVAE performs well on MCL-JCV and WebVid-Val in terms of PSNR, SSIm and LPIPS.

**Strengths:**

The experiments show ARVAE works well on two datasets compared with 6 recent video autoencoders.

**Weaknesses:**

The goal and motivation of this work are not that clear. The purpose of ARVAE is to help video generation or achieve better video compression? What are the compression ratio and efficiency?

The idea of performing motion estimation first and then encode the spatial difference frame-by-frame is not novel, which is the common practice for decades in video compression standards like MPEG or H.264/5 etc.

The technical presentations are not clear. What are the exact architectures of the temporal encoder and spatial encoder? What are the exact definition of the propagated features P_e? etc.

The paper claims ARVAE “without loss of information” (ll.098). Since there are motion estimator and many downsampling and upsampling operations, and the performance is measured by PSNR, please justify in what sense it is “without loss of information”.

**Questions:**

What are the exact architectures of motion extraction blocks and feature extraction blocks?  the same number of N blocks for both kinds of blocks? What is the exact warping operation? The warping is applied to the entire frame?

---

> ### Author Response · Authors · 2025-11-21
>
> We appreciate the reviewer’s feedback. We hope the following responses address your concerns.
>
> > W1 The goal and motivation of this work are not that clear
>
> **Answer to W1** Our ARVAE is a novel architecture for more efficient video compression and better video reconstruction quality, and latent space is suitable for video generation. Since this is a novel architectural design, this paper primarily focuses on demonstrating its effectiveness in video compression and reconstruction. As shown in Table 1, The compression ratio and efficiency have been validated in our experiments.
>
> > W2 The idea of performing motion estimation first and then encode the spatial difference frame-by-frame is not novel
>
> **Answer to W2** Codecs have long adopted spatiotemporal decouple idea, but in VAEs designed for video generative tasks, this idea has not been implemented. Furthermore, in conventional codecs, temporal information is encoded as coarse motion vectors formed by blocks of identical values, which facilitates direct reduction of motion data through redundancy. Nevertheless, this representation is unsuitable for feature extraction and compression with neural networks. By contrast, our approach employs dense optical flow to capture temporal information.
>
> > W3 What are the exact architectures of the temporal encoder and spatial encoder
>
> **Answer to W3** As described in 3.2.1, temporal encoder receives the motion field and the decoded previous frame to generate a feature.  As described in 3.2.2, spatial encoder receives the current frame and the feature generated by temporal encoder to generate the spatial supplement as part of the latent space. Propagated features are a group of multi-scale motion features. (As detailed in Fig. 3).
>
> >W4 Justify in what sense it is “without loss of information
>
> **Answer to W4** The idea here is that if we only separate a video into temporal and spatial components without applying compression, as shown in Fig.1, no information is lost. However, representing the video in these two parts facilitates compression. Once the temporal component is further compressed, information loss is inevitable.
>
> >Q1 What are the exact architectures of motion extraction blocks and feature extraction blocks?
>
> **Answer to Q1** The motion extraction blocks is consisted of Residual blocks and Convolutions blocks to extract features of the estimated motion (optical flow), and the feature extraction blocks extract features from the input RGB image. The number of blocks is the same for the same downsample ratio.
>
> >Q2 What is the exact warping operation. The warping is applied to the entire frame?
>
> **Answer to Q2** The warping operation displaces the pixels in an image according to the provided motion information. Yes, the warping is applied to the entire frame

---

### Official Review · Reviewer_VRHQ · 2025-11-01

**Soundness:** 3
**Presentation:** 2
**Contribution:** 2
**Rating:** 4
**Confidence:** 3

**Summary:**

This paper proposes a novel Autoregressive Video Autoencoder (ARVAE) that encodes and reconstructs video frames in an autoregressive manner. The key innovation lies in the decoupled temporal and spatial latent representation, where motion fields are used to model temporal consistency, and residual spatial information is encoded as a supplement for new content in each frame. The framework includes a temporal encoder/decoder, a spatial encoder/decoder, and a multi-stage training strategy that progressively increases sequence length to enhance temporal modeling. Extensive experiments demonstrate that ARVAE achieves state-of-the-art reconstruction quality with significantly fewer parameters and training data than existing baselines, and the method is applicable to both video reconstruction and generation tasks.

**Strengths:**

- Clear Motivation and Novelty:

The paper addresses a long-standing challenge in video autoencoding: entangled spatial-temporal modeling, and propose temporal-spatial decoupling is a conceptually clean and potentially impactful approach.

- Autoregressive Design:

Autoregressive frame prediction aligns well with the temporal nature of video data and allows for variable-length sequence modeling, which is an advantage over fixed-length VAEs.

**Weaknesses:**

- Lack of theoretical insight into key design choices:
The paper proposes a decoupled latent representation that separates temporal motion from spatial content. While this design is empirically effective, the underlying rationale is not well articulated. It remains unclear why this separation leads to better generalization or compression efficiency compared to joint modeling. A deeper conceptual or theoretical discussion would strengthen the paper’s contribution.


- Efficiency claims are insufficiently supported:
Although the method is described as lightweight and efficient, the paper lacks quantitative runtime analysis. Key metrics such as inference time, memory usage, and decoding FLOPs are not reported. Without such data, it is hard to objectively validate the claimed efficiency advantages over existing models.

- Experimental design lacks clarity in some parts:
While the authors conduct both qualitative and quantitative comparisons, the presentation of results could be more focused. In particular, the visual comparisons do not clearly highlight where ARVAE outperforms other methods (e.g., handling of motion, texture fidelity, or temporal consistency). Similarly, the numerical results, though comprehensive, are not always clearly interpreted. More targeted analysis would help clarify the distinctive benefits of the proposed method.

- Ablation studies could be more thorough:
The ablation experiments successfully demonstrate the importance of individual components (e.g., multi-scale features, state features, and multi-stage training). However, the analysis does not explore potential failure cases or limitations. For example, it would be helpful to understand how the model behaves under challenging conditions such as fast motion, occlusion, or long-range propagation. This would provide a more complete picture of the method’s strengths and weaknesses.

- Presentation and clarity:
Some figures—particularly the architecture diagrams—lack sufficient detail or clear labeling, which may hinder understanding for readers unfamiliar with the topic. Additionally, certain sections in the methodology could benefit from more formal definitions and clearer notation to improve technical clarity.

**Questions:**

See weakness.

---

> ### Author Response · Authors · 2025-11-21
>
> We appreciate the reviewer’s feedback. We hope the following responses address your concerns.
>
> > W1 Lack of theoretical insight into key design choices
>
> **Answer to W1** Regarding compression efficiency, we explained in the introduction and Figure.1 that our spatiotemporal separation method produces latent representations with lower entropy after separation, thereby allowing greater compression of video data. For better generalization, we will explore it in future work by more generation experiments, but this work primarily focuses on video reconstruction quality.
>
> > W2 Efficiency claims are insufficiently supported
>
> **Answer to W2**  In the appendix Table 7, we provide a comparison of GFLOPs across different models.
>
> > W3 Experimental design lacks clarity in some parts
>
> **Answer to W3** As shown in Figure 4, our method achieves superior and clearer reconstruction of the cartoon character’s eyes, tongue and the fork. Quantitatively, our method matches the baseline performance at 8X downsampling, and achieves better reconstruction performance at 16X and 32X downsampling.
>
> > W4 Ablation studies could be more thorough
>
> **Answer to W4** As shown in Table 5, we provide the reconstruct quality across different video length. More visualization results is shown in Figure 5 and Figure 6 which shows that ARVAE achieves superior reconstruction quality in image details.
>
> > W4 Presentation and clarity
>
> **Answer to W5** Sorry for the unclarity, we will revisit the paper for better presentation

---

### Official Review · Reviewer_p2We · 2025-11-02

**Soundness:** 2
**Presentation:** 2
**Contribution:** 2
**Rating:** 2
**Confidence:** 4

**Summary:**

The manuscript presents a new video autoencoder ARVAE, aiming to address difficulty in capturing the temporal consistency of existing video autoencoders. The video frames are compressed and reconstructed in a frame by frame manner in ARVAE, such that a flexible number of frames can be processed. For each frame, ARVAE converts it into decoupled spatial and motion representations using a two-branched model, and the decoder reconstruct the image based on the decoupled representations. Emprical results show that ARVAE is highly parameter efficient, using 0.1M parameter to achieve similar performances to video autoencoders with more than 10M parameters.

**Strengths:**

1. ARVAE is motivated to address the temporal consistency in video generation, through a decoupled video representation composed of spatial and motion components. Intuitively speaking, this strategy is capable of preserving spatial details of previous frames.

2. ARVAE shows strong parameter efficiency, achieving strong reconstruction performance compared to video autoencoders with 100x larger parameter scale.

3. Ablation experiments show the effectiveness of using multi-scale propagated features and reconstruction with state features.

**Weaknesses:**

1. Though reconstruction is an important evalaution aspect for autoencoders, it is even more important to evaluate whether the latent is good for further generation process. Yet the manuscript provides limited evaluation on the generation side. First, since the representations are decoupled, it is unclear how the spatial and motion representations are used during the generation process. Second, the comparison is limited to the comparison between system-level generation results, with no controlled experiments on whether the learned representation is easy or difficult to generate. This usually requires using a standard generation model, VSD for example, and compare different latents under the same training data and number of iterations.

2. It seems to me that the ARVAE could not achieve temporal compression. That is to say the temporal compression ratio is 1, which is probably the underlying reason why the model is able to achieve strong reconstruction performance with such small parameter scale. The temporal compression ratio also needs to be explicitly specified in the comparison. It would be better if FLOPs comparison could be included as well.

3. Though the manuscript is motivated to improve temporal consistency, the experiments fail to show that ARVAE is capable of improving the temporal consistency of reconstructed or generated videos. Though qualitative results are provided for video reconstruction, the difference between different autoencoders is negligible.

**Questions:**

- Is spatial feature and temporal feature concatenated for video generation?

---

> ### Author Response · Authors · 2025-11-21
>
> We appreciate the reviewer’s feedback. We hope the following responses address your concerns.
>
> > W1 It is more important to evaluate whether the latent is good for further generation process.
>
> **Answer to W1** Since we propose a novel design approach for video VAE architectures, this paper—similar to other works such as VideoVaePlus[1], Vidtok[2], Vidtwin[3]—primarily focuses on video reconstruction quality. The effectiveness of the generative component will be the focus of future work. In video generation, spatial representations are pre-provided images, such as a generated picture or a real image; motion representations are the target latent space that the generative model generate conditioned on the spatial representations. Once the motion representations are generated, the VAE decoder combines spatial and motion representations to reconstruct the complete video.
>
> > W2 Temporal compression.
>
> **Answer to W2** Our method does not perform temporal compression, but our latent representation has fewer channels, so the comparison with previous methods is fair. Specifically, at 8× and 16× downsampling, our model uses 1/4 the number of channels compared to the baseline, while the baseline applies 4× temporal compression. As a result, the latent sizes of both methods are equivalent. At 32X downsampling, LTX uses 128 channels whereas we use only 16, so our latent representation is even smaller than LTX.
>
> > W3 Improving the temporal consistency.
>
> **Answer to W3** This paper primarily focuses on introducing a autoregressive VAE architecture that achieves both efficiency and superior reconstruction quality. Moreover, the generative results on the Taichi and Sky datasets can to some extent demonstrate the effectiveness of our ARVAE's performance on video generation, and we will continue work on the generation part in the future.
>
>
> [1] Xing, Yazhou, et al. "Large motion video autoencoding with cross-modal video vae." arXiv preprint arXiv:2412.17805 (2024).
>
> [2] Tang, Anni, et al. "Vidtok: A versatile and open-source video tokenizer." arXiv preprint arXiv:2412.13061 (2024).
>
> [3] Wang, Yuchi, et al. "Vidtwin: Video vae with decoupled structure and dynamics." Proceedings of the Computer Vision and Pattern Recognition Conference. 2025.

---

### Official Review · Reviewer_BNeC · 2025-11-02

**Soundness:** 3
**Presentation:** 2
**Contribution:** 2
**Rating:** 4
**Confidence:** 5

**Summary:**

This paper proposes an Autoregressive Video Autoencoder (ARVAE) for video reconstruction. The core idea is to encode and decode each frame autoregressively, conditioned on its predecessor. The proposed latent representation is decoupled into a temporal motion component, derived from an estimated optical flow, and a spatial supplement component, representing the residual information. The authors claim that this approach achieves superior reconstruction quality with a significantly smaller model and less training data compared to state-of-the-art methods that perform joint spatiotemporal compression. Experiments are conducted on standard benchmarks, and an application to video generation is briefly explored.

**Strengths:**

1. The primary strength of this paper lies in its reported quantitative results. As shown in Table 1, the proposed ARVAE achieves state-of-the-art or highly competitive performance on PSNR, SSIM, and LPIPS metrics across multiple benchmarks, while claiming to use a fraction of the model parameters and training data of strong baselines like Step-Video and LTX-Video. If these results are validated to be under fair comparison, they would be quite significant.
2. The autoregressive approach, which leverages the previous frame to reconstruct the current one, is conceptually simple and intuitive. It builds upon a long-standing principle in classical video coding (i.e., predictive frames), making the method easy to understand.

**Weaknesses:**

[Poorly Justified Motivation] The paper's motivation is weak and questionable. The authors claim that existing methods "model the video with spatial and temporal information intermingled, failing to effectively capture the temporal consistency" (Lines 41-42), and they single out "3D attention mechanisms" as the target of criticism. This is problematic for two main reasons: First, 3D attention is not a dominant component in most state-of-the-art video VAEs; 3D convolutions are far more prevalent. Attacking a niche design choice makes the motivation feel misplaced. Second, the claim that 3D convolution leads to harmful spatiotemporal entanglement is not a community consensus. The authors provide no theoretical analysis, empirical evidence, or citations to support this assertion, making the premise of the work feel unsubstantiated.

[Crucial Methodological Details are Missing] The paper omits critical details necessary for a fair assessment of the work's contributions. The "off-the-shelf image-based autoencoder" used for the first frame is never specified. The quality of this "Img-frame" is fundamental to the entire autoregressive chain, and its architecture and performance are unknown. The "motion estimator" is treated as a black box. The paper does not specify the network architecture (e.g., PWC-Net, RAFT).

[Questionable Experimental Choices and Result Quality]
The model is not trained with a GAN loss, which typically leads to blurry reconstruction results for VAEs. The authors seem to have avoided this issue by presenting very few qualitative examples (only 4 images in the paper), three of which are cartoons, and one is inherently blurry. This selective visualization feels like cherry-picking and fails to convincingly demonstrate the model's ability to reconstruct sharp details in realistic, complex scenes.

[Misleading or Unfair Comparisons in Experiments] The experimental results, while seemingly impressive, are presented in a misleading way that conceals major disadvantages.
1.	Unfair Compression Ratio: The proposed method does not perform temporal compression (the temporal compression ratio is 1x), whereas the SOTA methods it compares against (e.g., Step-Video, LTX-Video) employ significant temporal compression (e.g., 4x or 8x). This is a massive, unfair advantage that is not highlighted in the tables, rendering the numerical comparison of reconstruction quality largely meaningless.
2.	Impractical for Downstream Tasks: The lack of temporal compression means the number of latent tokens for a video clip would be extremely large (4x or 8x more than competitors). This would make training a downstream diffusion model computationally prohibitive, questioning the practical utility of the proposed VAE.
3.	Inaccurate Parameter Count: The claim of a tiny 6M parameter model is highly suspect. The model consists of an off-the-shelf image autoencoder, a motion estimator, and the ARVAE encoder/decoder. This reported number appears to be incorrect or at least incomplete, potentially omitting the parameters of the motion estimator and the first-frame encoder.
4.	Poor Temporal Generalization: Table 5 shows that performance degrades as the sequence length increases, even for a short 16-frame clip. This is a drawback of autoregressive models, and the paper fails to evaluate the model on longer sequences to assess the critical issue of error accumulation.
5.	Outdated Generation Baselines: The video generation experiment (Table 6) compares against GAN-based methods from 2022. This comparison is meaningless in the current landscape dominated by diffusion models.
6.	Missing Inference Time Analysis: The paper completely omits any discussion of inference speed. I suspect the author's motion estimator is a flow model, but flow models like RAFT require an extremely high number of iterations to produce accurate optical flow and are exceptionally slow. Yet the author avoids any relevant description of this.

In summary, the paper appears to selectively present results and comparisons in a way that is favorable to itself while omitting critical details and fair evaluations. The motivation is weak, the method description is incomplete, and the experiments are fundamentally flawed in their setup and reporting. Therefore, I cannot recommend acceptance.

**Questions:**

See weaknesses

---

> ### Author Response · Authors · 2025-11-21
>
> We appreciate the reviewer’s feedback. We hope the following responses address your concerns.
>
> > W1 Poorly Justified Motivation
>
> **Answer to W1** : For the motivation of the paper, if we feed an entire video directly into a neural network, the network is essentially processing both temporal and spatial information of the video simultaneously. While this approach is not necessarily harmful, we believe it is inefficient than our proposed decouple based method. As shown in Figure 1, the shannon entropy of S-T decoupled representation is minimum, indicating that this representation has greater compression potential. Moreover, in codecs such as H.264, it is a common consensus that separating temporal and spatial information (e.g., motion vectors and residuals) before compression, and it leads to higher compression efficiency. However, in VAEs designed for generative tasks, we have not yet seen this kind of decoupled representation design. Moreover, sorry for the unclear explanation, we don't attack the design choice of "3D attention" or "3D convolutions", we just state that current methods extend 2D image VAE to 3D video VAE by those operations.
>
> > W2 Crucial Methodological Details are Missing
>
> **Answer to W2** : Sorry for the unclear explanation, we chose Flux’s VAE as the image encoder. The architecture of  motion estimator is Spynet[1], and we random initialized the parameter of the motion estimator for end to end training.
>
> > W3 Questionable Experimental Choices and Result Quality
>
> **Answer to W3** : According to our understanding, adding a GAN loss can improve the reconstruction quality of a VAE, but it reduces the generalization ability of the latent representation (leading to poorer performance on downstream understanding tasks). In contrast, our method achieves performance comparable to previous approaches even without incorporating a GAN loss, which suggests greater potential for further improvement. For the visualization results, those video clip is also used in VideoVAEPlus[2]. In particular, the motion amplitude in cartoon video segments is relatively large, and other models do not perform well on them.
>
> > W4 Misleading or Unfair Comparisons in Experiments
>
> **Answer to W4** :
> 1. The comparison of different methods is fair, because the latent representation in our method has fewer channels than previous methods. At 8X and 16X downsampling, our model uses 1/4 the number of channels compared to the baseline, while the baseline applies 4X temporal compression. As a result, the latent sizes of both methods are equivalent. At 32X downsampling, LTX uses 128 channels whereas we use only 16, so our latent representation is even smaller than LTX.
>
> 2. For generation tasks, compression along the channel dimension and compression along the time dimension are equivalent in improving the efficiency of generation tasks.
>
> 3. We apologies for the unclear explanation. The image vae we use is Flux’s VAE which contains 83M parameters. The parameters we report is only our ARVAE, which includes the motion estimator and the ARVAE encoder/decoder. The motion estimator (SPyNet) is a very small network, with only 0.96M parameters. For the first frame, we use the 83M FLUX VAE to compress, while for subsequent frames in temporal propagation, each frame is processed with a 6M ARVAE. Taking the experiment of reconstructing 16 frames as an example—the first frame with the Flux VAE plus 15 frames with the ARVAE—the average number of parameters to compress one frame is 11.38M.
>
> 4. The issue of declining reconstruction quality in long videos is indeed present; however, in practical applications, one may consider generating the video in segments, ensuring that the initial frame of each segment maintains a high quality.
>
> 5. Our work aims to propose an efficient Video AE for video compression and reconstruction. Performing generation tasks is an extension of our method to downstream applications.  Since our training resources are limited, we can't retrain large video generation models, and we chose to train the generative model on small datasets to provide a reference.
>
> 6. In the appendix Table 7, we provide a comparison of GFLOPs across different models. Below we provide the GFLOPs on one 224x224 image and parameters of the flow model we use. As shown in the results, the flow model is a lightweight model.
>
> |    Model    | GFLOPs | #Params   |
> |-------------|------------|-------------|
> | flow model| 15.9        | 0.96M       |
> | LTX| 82.1     | 419M       |
>
>
> [1]Ranjan, Anurag, and Michael J. Black. "Optical flow estimation using a spatial pyramid network." Proceedings of the IEEE conference on computer vision and pattern recognition. 2017.
>
> [2]Xing, Yazhou, et al. "Large motion video autoencoding with cross-modal video vae." arXiv preprint arXiv:2412.17805 (2024).

---

### Meta-Review · Area_Chair_zrgE · 2026-01-07

**Summary:**

The paper proposes an Autoregressive Video Autoencoder to improve video reconstruction through a decoupled representation of temporal and spatial information. The authors report competitive reconstruction quality with fewer parameters, but the motivation for the approach is weak, as pointed out by reviewers. Some main methodological details are missing, particularly regarding the architectures used and the motion estimation process. The experimental design raises concerns, including comparisons with state-of-the-art methods that employ temporal compression. Also, the paper provides limited qualitative evaluations, so the reviewers have doubts about the model's performance in real-world scenarios. Overall, the paper does not meet the standards for acceptance due to its methodological shortcomings and lack of clarity. Thus, the AC recommends rejection.

**Reviewer Concerns:**

The author's response has addressed some concerns raised by the reviewers, regarding the clarity of the architecture used for the temporal and spatial encoders, as well as the motivation behind the decoupled representation. The authors provided details on the motion estimator and clarified how their approach differs from traditional video compression methods.

However, some concerns remain unaddressed. The lack of theoretical insight into the benefits of the decoupled representation compared to joint modeling is not convincing. The paper's experimental design remains questionable, particularly regarding the comparisons with other methods that utilize temporal compression, which ARVAE does not. Furthermore, the rebuttal did not sufficiently address the limited qualitative evaluations or the implications of not using a GAN loss, which could affect the ability to generate sharp and realistic video reconstructions.

**Reviewer Scores:**

This paper has received ratings of 2,2,4,4 in the end. I don't think reviewers would change their scores.

---

### Decision · Program_Chairs · 2026-01-26

Reject